# Mainline Railway Modeled with 2100 MHz 5G-R Channel Based on Measured Data of Test Line of Loop Railway

**Yiqun Liang** [1,2,3] , **Hui Li** [2,3,*], **Yi Li** [2,3] and **Anning Li** [2,3]

1. Postgraduate Department, China Academy of Railway Sciences, Beijing 100081, China; liangyiqun@139.com
2. Signal and Communication Research Institute, China Academy of Railway Sciences Corporation Limited, Beijing 100081, China; liyi_swjtu@foxmail.com (Y.L.); 18801307125@163.com (A.L.)
3. National Research Center of Railway Intelligence Transportation System Engineering Technology, Beijing 100081, China
* Correspondence: tkthlh@sina.com; Tel.: +86-139-1168-6560

**Abstract:** Railway communication is undergoing a transitional period worldwide, and 5G for railway (5G-R), which has been adopted by the Railway International Union (UIC) in China and in many other countries, which is a common concern all over the world. The 2100 MHz frequency band, which ranks among the top three mainstream frequency bands for public 5G network deployment globally, has recently been issued by MIIT for a 5G-R trial in China, and the frequency is fairly close to 5G-R frequency band of 1900 MHz in Europe. Propagation study for railway communication, especially for mainline, which is the most common and widely used scenario, is of great importance, because it is closely related to passengers' lives and properties. In this article, based on the 5G-R dedicated network testing environment constructed along the test line of loop railway of National Railway Track Test Center of China, the 2100 MHz 5G-R propagation measurement campaign is conducted, with an in-service SS-RSRP signal-based passive 2100 MHz large-scale channel testing system, as well as a SDR-based 2100 MHz 5G-R channel sounding system, based on which the large-scale as well as small-scale channel characteristics are derived and extensively analyzed. With data derived from the passive testing system, the classical FI, CI as well as TR 38.901 large-scale channel models are properly fitted and compared by calculating RMSE as well as MAE values. Moreover, based on the SDR-based channel sounding system, conclusions on the relationship between the parameters of multi-path component numbers, RMS time spread, received power and distance between the transmitter and the receiver are drawn. The 5G-R system studied in this letter is symmetric FDD communication system, apart from the 2 aspects of channel modeling characteristics, which are large-scale and small-scale complementing each other, the research methodology adopted of the letter also has two aspects, passive measurement, as well as active measurement, and both methods have their own advantages and different focuses, which can also complements each other. Relevant research results of this letter will be helpful for facilitating the R&D, deployment, as well as network optimization of a future mobile communication system under railway mainline scenario.

**Keywords:** 5G-R; 2100 MHz; railway; mainline; passive; SS-RSRP; SDR; channel modeling; measurement; test line of loop railway; MPCs; RMS delay spread; K-factor

## 1. Introduction

Communication technology has brought unprecedented convenience in information transmission and acquisition. Symmetry plays an important role in the field of communication engineering. Channel symmetry between uplink and downlink as well as frequency division duplexing communication systems have been fully utilized to date. Symmetric algorithm, topology and connection have been widely applied in communication networks. With the rapid increase in communication requirements, there are some challenging issues for symmetric FDD communications, which is the case of this article. Large-scale and small-scale channel characteristics are two important aspects of channel modeling. The former is

closely related to the interval of the deployment spacing of base stations along the railway. When base stations are densely deployed, although the signal coverage level (SS-RSRP) can be guaranteed, it can also lead to severe interference between 5G-R cells, as well as high costs for base station equipment, towers, station buildings, supporting transmission and power equipment, infrastructure, land acquisition, engineering construction and other issues. When the deployment distance of the base station is too large, it will cause the received signal level of the train to be low when it reaches the edge of the cell, resulting in the inability to guarantee the signal-to-noise ratio (SS-SINR), leading to a low transmission rate and the poor connection reliability of 5G-R carrying services. As for the small-scale characteristics, the high-speed movement of trains leads to small-scale channel propagation problems such as the rapid fading of wireless channels and severe Doppler spread, resulting in a significant decline in the performance of railway 5G-R railway dedicated communication systems, directly affecting the transmission rate and reliability of railway communication applications carried by 5G-R.

Railway radio communication, as the foundation of promoting efficiency as well as reliability of railway transportation, reliably carries key services such as dispatching communication, train control information, maintenance communication, etc, and is closely related to the safety of passengers' lives. Since frequency resource is the physical base and premise of railway radio communication, to allocate and protect dedicated frequency resources to railway radio communication system is a common consensus, from the point of view of the International Union of Railway as well as many non-European countries.

The leading global system of railway dedicated mobile communication (GSM-R), using railway dedicated frequency mentioned above, has been a great success, which has covered 163,000 km of railway lines in Europe and 90,000 km of railway lines in China. However, GSM-R, as a narrow band system, is unable to meet the constantly evolving demand for train-ground transmission bandwidth. Future Railway Mobile Communication System (FRMCS) is a project of International Railway Union (UIC) which aims to research and develop the successor of GSM-R, and 5G-R was chosen by FRMCS as the future mobile communication standard for railway; in the meanwhile, however, the 5G band 1900–1910 MHz, which is defined as n101 by 3GPP, is allocated to the 5G-R system in Europe. China Railway also adopted 5G-R for the new generation mobile communication, and in September 2023, the Ministry of Industry and Information Technology of China (MIIT) issued a document permitting China Railway to conduct tests on a test line of loop railway located in the northeast part of Beijing, China, using a frequency of 2100 MHz (1965–1975 MHz/2155–2165 MHz, FDD), defined as n1 by 3GPP, which is relatively close to that of Europe. According to a survey conducted by GSMA, the 2100 MHz frequency band ranks among the top three most frequently adopted frequency bands, second only to the 3500 MHz and 700 MHz frequency bands in terms of 5G deployment, as seen in Figure 1.

The networking of a railway mainline mobile communication system is narrow-strip-shaped, which means that base stations are deployed along the railway line. Since railways are ramified all over the country, operating scenarios have to be considered, including viaducts, cuttings, stations, hilly terrain, as well as open space [1]. Apart from special propagation scenarios, a high moving speed over 350 km/h, a higher quality of service (QoS), severe electromagnetic environments and interference are challenges faced by railway-dedicated mobile communication systems [2]. Moreover, the non-stationary characteristics of railway channel, which belong to short-term fading behavior, are closely related to dynamic channel modeling, especially for HSR [3]. All in all, railway radio channel models describe how channel fading behaves in a given scenario, and helps design communication systems and evaluate link- and system-level performance, and the study of channel modeling for railway communication is of great importance.

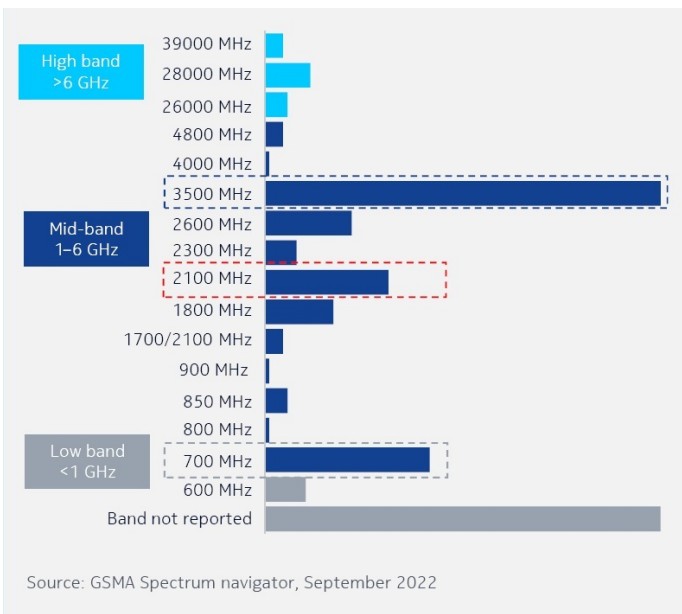

**Figure 1.** Frequency band distribution of 5G network.

### 1.1. Related Works

Scholars have conducted a series of related work with respect to the main railway line channel modeling. Related literature will be reviewed from four aspects.

From the point of view of a communication scenario, different terrains such as railway tunnels, viaducts, cuttings, urban areas, rural areas, as well as different communications scenarios such as train-to-train communication, in-train communication, air-to-ground communication, dynamic channel model with overhead line poles, are all covered by open literature. In [4], the tunnel entrance scenario channel time correlation function is analyzed based on measurement data. In [5], based on measurements under different viaduct scenarios at 930 MHz conducted along the "Zhengzhou–Xian" HSR in China, the authors analyzed the impacts of the viaduct height as well as the number of surrounding scatterers on the channel characteristics. In [6], the authors conducted measurements in the cutting scenario along the "Zhengzhou–Xian" HSR, based on which the scattering components are modeled by clusters.The authors in [7] focused on the short-term fading behavior of HSR channel with multiple scenarios, such as the rural, station, as well as suburban. In [8], the train-to-train scenarios are studied, considering typical environments. In [9], the authors studied the indoor wireless channels in HSTs, and the specific interior layout of the train is considered. In [10], the air-to-ground channel at 1.4 GHz is discussed, and the altitude of the transmitter is 20 km or 100 km, and the receiver is placed on the train. In [11], the authors proposed the impact of the overhead line poles—which constitute the infrastructure of the power supply system built along the railway line—on the line-of-sight path of railway channel propagation.

Talking about the frequency bands, the higher the frequency, the greater the path loss, and as a broad consensus, countries around the world generally allocate medium- and low-frequency resources for railway mainlines. As a result, academic research into the frequency characteristics of the railway main lines mostly focus on sub 6 GHz. Ref. [12] analyzed the performance of the Indonesia Railway Channel Model, with a frequency of 873–880 MHz and 918–925 MHz for up-link and down-link, respectively. In [13], based on measurements in railway scenarios at a frequency of 930 MHz, the authors presented a general stochastic fading channel model. The authors in [14] created a new 1.905 GHz channel model for HSTs derived from a measurement campaign conducted on the Beijing–Tianjin commercial HST in China. In [15], measurements of railway cuttings at 950 MHz and 2150 MHz were carried out, and the broadband measurement results were analyzed. In [16], measurements along the Harbin–Dalian passenger dedicated railway line of China were conducted at

2.6 GHz with 40 MHz bandwidth, based on which the single-input–single-output (SISO) channel characteristics were derived. In [17], the authors conducted measurements at 2.7 GHz and 5.6 GHz on a German high-speed track with train speed of 300 km/h, and CIR, the coherence bandwidth, as well as RMS delay are deeply analyzed. Ref. [18] focused on 3.5 GHz channel characteristics for 5G in urban rail station.

Moving on to research methodology, measure campaign, ray-tracing simulations, machine-learning, deep learning, signal-to-noise ratio quantization strategy, finite-state Markov modeling, multi-task learning, as well as passive channel sounding all have been studied by scholars. In [19], the wireless channels fading statistics of HSR cutting scenarios are characterized by conducting measurements, and described by a hidden Markov mode. In [20], the tapped delay line (TDL) models of ray-tracing simulations are used to evaluate the channel state as well as the throughput of 5G wireless communication systems in various HSR scenarios. In [21], machine learning approaches were used to solve the problem of wireless channel scenarios identification. In [22], machine learning (ML) was adopted to investigate the MPCs clustering in typical HSR scenarios. In [23], a deep learning (DL) method was used to analyzed the spatial-temporal prediction of channel state information (CSI) as well as channel statistical characteristics (CSCs) for the future smart HSR communication network. In [24], the signal-to-noise ratio (SNR) threshold was exploited by authors to established a novel finite-state Markov chain (FSMC) optimization simulation model, by which, the universality and accuracy of channel models can be improved in different HSR scenarios. In [25], the small-scale fading channels of the "Luoyang South–Lintong East" HSR in China were characterized via the finite-state Markov channel (FSMC) modeling. In [26], a multi-task learning (MTL) convolutional neural network (CNN) was adopted by the authors to propose a novel super-resolution (SR) model for generating channel characteristics data. In [27], a passive channel sounder was used to extract 3.45 GHz 5G network's channel impulse responses (CIRs).

In terms of research object, except channel parameters such as power delay profiles (PDPs), path loss, Rician K-factor, RMS delay spread, re-configurable intelligent surface (RIS), multi-link channels, narrow-beam channel, Doppler shift, as well as propagation scenario identification, all have been involved. In [28], the channel characteristics of RIS-assisted near-field communication was extensively analyzed, and a 3D RIS-assisted MIMO channel model was proposed. In [29], a Markov-based multi-link tapped-delay line model for railway multi-link transmission communications is established. In [30], common and uncommon clusters for the different links of HSR narrow-beam channels are described with a non-stationary 3D wide-band geometry-based stochastic model. Ref. [31] focused on the influence of mobility of train in rapidly time-varying channels. In [32], a new propagation scenario identification model was proposed using the long short-term memory (LSTM) neural network.

### 1.2. Contribution and Organization of the Paper

Through a literature review conducted in Section 1.1, we can tell that there is still a lack of evaluation of the 2100 MHz 5G-R channel modeling for the railway mainline based on measured data via the passive as well as active SDR-based measurement methods. The passive measurement method can realize the measurement of 5G-R networks under network operating conditions, and the construction of the measurement system is relatively simple. The transmitting end uses 5G-R base station signals, and only the on-board receiving end measurement equipment needs to be constructed to achieve the real-time collection of SS-RSRP as well as railway position information. This passive measurement method is more closely related to the real application scenarios of 5G-R systems, and more accurately reflects the power perceived and received by mobile terminals. It is not only suitable for using a single RRU device to build a station site for auxiliary site selection in the early stage of 5G-R system construction, but also suitable for the evaluation and optimization of 5G-R network level coverage levels in the stages of dynamic detection, daily inspection, network optimization and other operation and maintenance stages after

the completion of deployment of 5G-R network. While using the SDR-based method, more parameters and detailed information of the railway-dedicated communication channel can be derived, for instance, the MPCs, the received power and the delay of each path, and these parameters and information can be further used to disclose the characteristics of the channel. The novelty of this work is to construct a 5G-R dedicated network testing environment, and conduct a measurement campaign with an in-service SS-RSRP signal-based passive large-scale channel testing system, as well as an SDR-based channel sounding system, based on which, large-scale as well as small-scale channel characteristics are derived and deeply analyzed. The FI, CI as well as TR 38.901 large-scale channel models are properly fitted under the railway mainline scenario, with low RMSE as well as MAE values; meanwhile, the numerical results of relationship between parameters of MPCs, received power, RMS time spread and the distance between transceivers are drawn.

The rest of this article is arranged as follows.

- Section 2: Description of the 5G-R-dedicated network constructed along the test line of the loop railway, the location of the base stations as well as the characteristics of the antenna transceivers are disclosed;
- Section 3: The in-service SS-RSRP signal-based passive large-scale channel testing system and SDR-based channel modeling system, as well as the corresponding measurement and data processing methods, are presented;
- Section 4: The results are summarized;
- Section 5: Conclusions and future works are pointed out.

## 2. The 5G-R Testing Environment of Loop Railway Test Line

Our measurement campaign was conducted in National Railway Track Test Center, located in the northeastern part of Beijing, China, the test line of loop railway of which is the largest loop test railway line of Asia, with a length of approximately 9 km, as is shown in Figure 2.

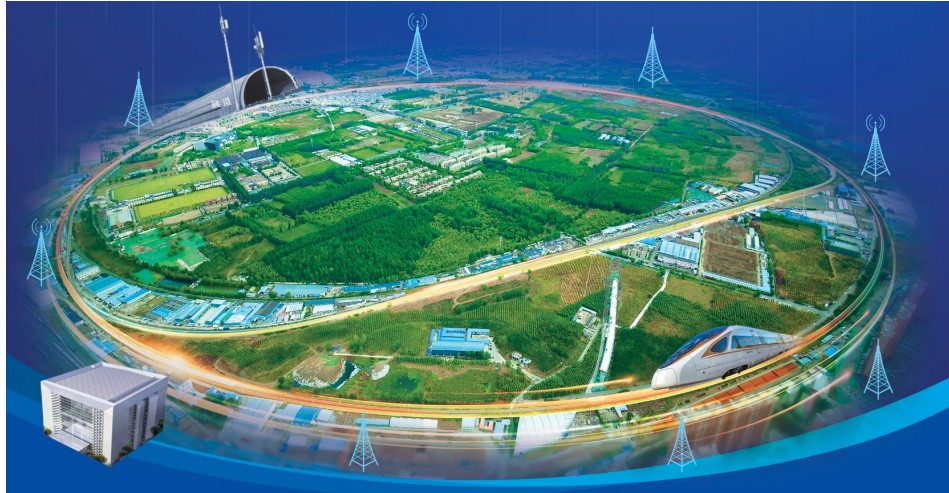

**Figure 2.** Test line of loop railway, National Railway Track Test Center.

A total of five 5G-R-dedicated network base stations were built along the loop, equipped with base station equipment from multiple manufacturers, to achieve full coverage of the 5G-R dedicated network, supporting railway characteristics such as strip-shaped coverage, redundant networking; moreover, the frequency band is 2100 MHz (1965–1975 MHz/2155–2165 MHz, FDD), defined as n1 by 3GPP, as mentioned in the introduction session. Locations of on-site equipment rooms and base station towers are shown in Figure 3, the locations of the five 5G-R base stations are K0.5, K1.1, K4.2, K6.0 and K7.1, respectively.

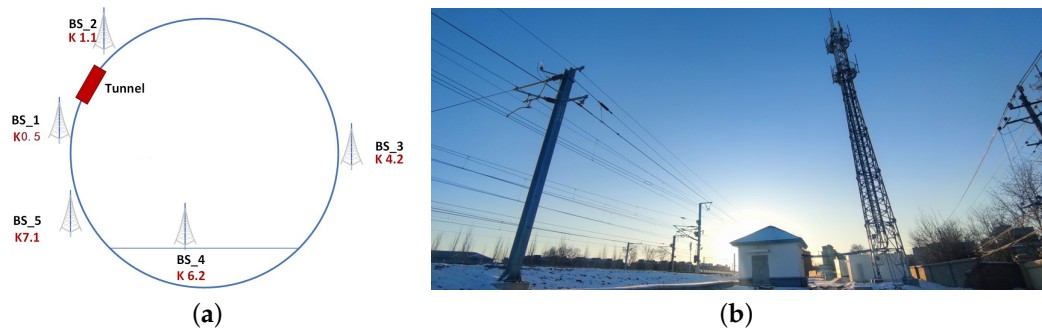

|  |  |
|---|---|
| (**a**) | (**b**) |

**Figure 3.** Locations of on-site equipment rooms and base station towers. (**a**) Locations of 5G-R Base Stations. (**b**) On-site equipment room and base station tower.

From the perspective of engineering installations, there are two installation methods for 5G-R RRU equipment.

As for the first installation method, 5G-R RRU equipment is mounted on the rack installed in the on-site equipment room, which in this case has the location of K1.1. The base station antenna is mounted on the second platform of the tower, with a height of 26 m from the rail track surface. The length of the feeder between is 50 m, with a feeder loss of 4.1 dB. The 5G-R RRU equipment, feeder and antenna are shown in Figure 4, in this scenario, although the feeder loss between the RRU and the BS antenna is much larger than the second scenario; however, the maintenance of RRU equipment is more convenient.

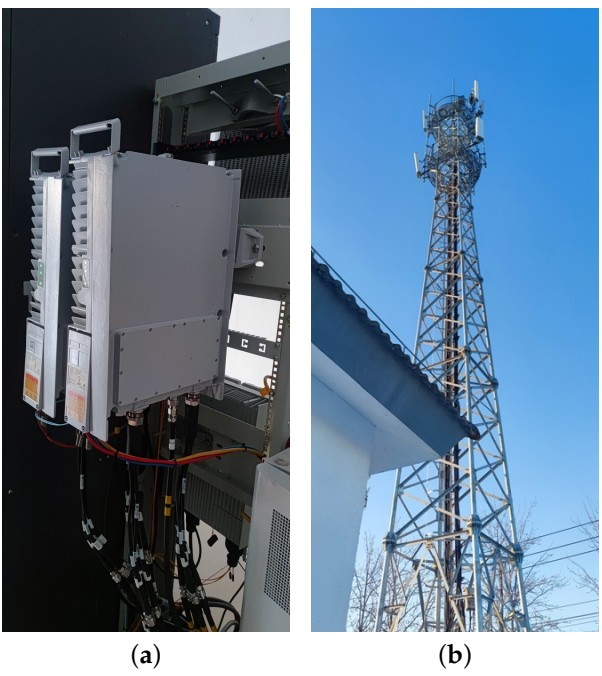

|  |  |
|---|---|
| (**a**) | (**b**) |

**Figure 4.** 5G-R RRU equipment and antenna. (**a**) 5G-R RRU equipment. (**b**) Base station feeder and antenna.

As for the other installation method, 5G-R RRU equipment is mounted on the tower, adjacent to the BS antenna, which is the case of location K4.2; as shown in Figure 5, in this scenario, the feeder loss between the RRU equipment and the antenna is very small, even negligible.

The 5G-R base station adopts 8T8R antenna, which has eight elements, $\pm 45°$ polarization, the horizontal and vertical antenna directional patterns are shown in Figure 6.

The on-board roof antenna is mounted on the roof of the testing locomotive, with a spacing of 1.5 m between adjacent antennas to avoid mutual interference, as shown in Figure 7, and the roof antenna is 4.2 m above the rail track surface.

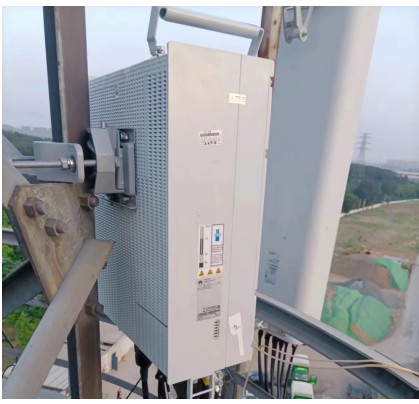

**Figure 5.** RRU installed on the tower, adjacent to the antenna.

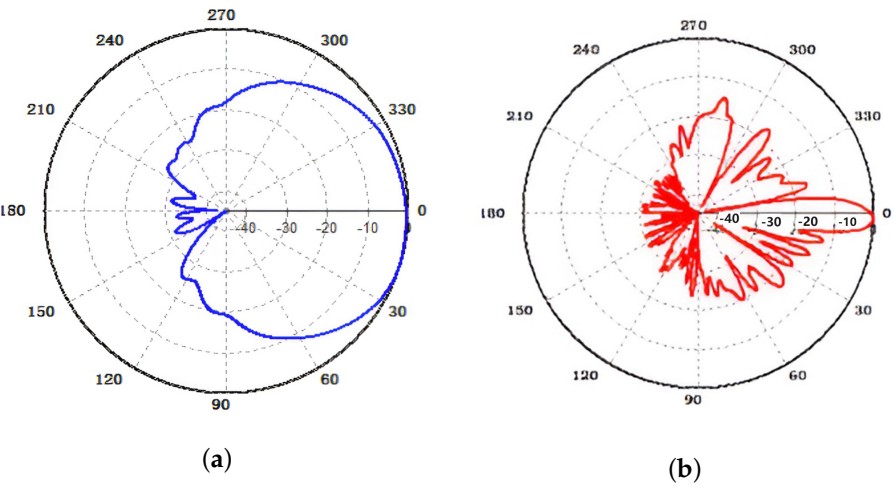

(**a**)  (**b**)

**Figure 6.** 5G-R base station antenna pattern. (**a**) The horizontal directional antenna pattern. (**b**) The vertical directional antenna pattern.

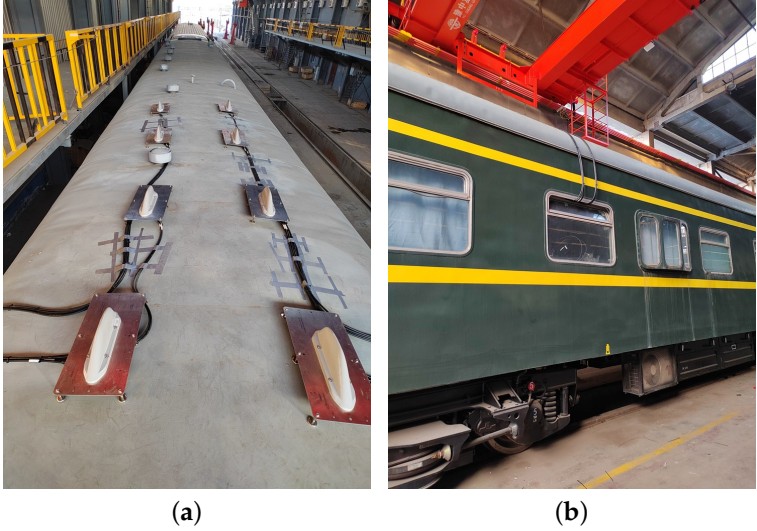

(**a**)  (**b**)

**Figure 7.** Testing locomotive and roof antenna. (**a**) Roof antenna. (**b**) Feeder installation.

The horizontal and vertical antenna directional patterns of the on-board roof antenna are shown in Figure 8.

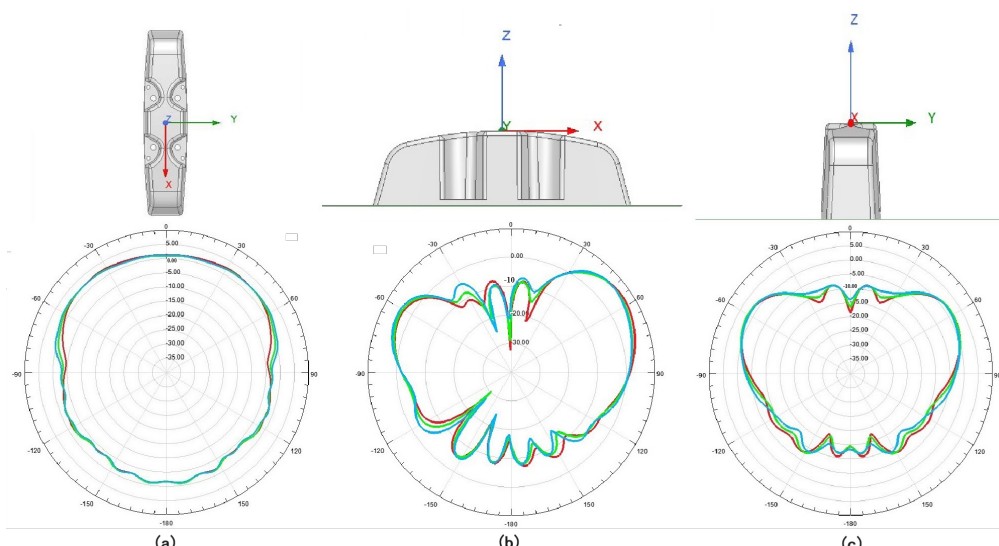

**Figure 8.** On–board roof antenna pattern. (**a**) The X–Y plane antenna pattern. (**b**) The X–Z plane antenna pattern. (**c**) The Y–Z plane antenna pattern.

The terrain of our test environment can be classified as a rural area, with trees and sparse low buildings around; moreover, specific railway objects are present in our scenario, such as the low partition wall along the railway, as well as catenary poles, as shown in Figure 9.

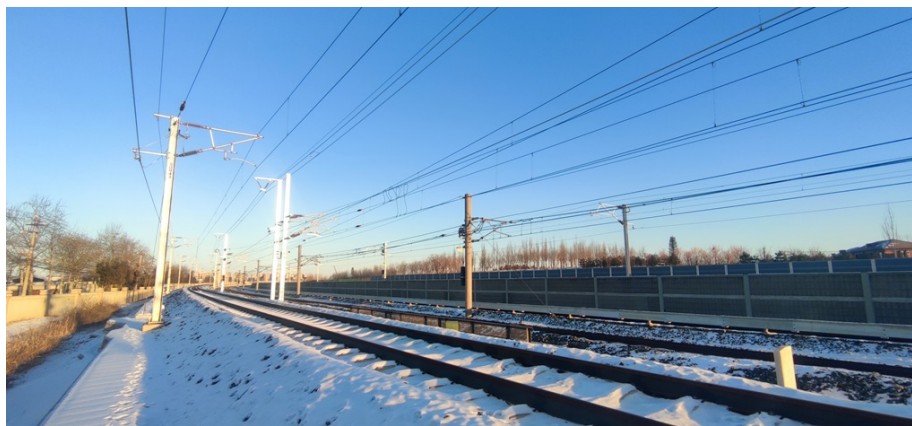

**Figure 9.** Terrain of the test environment.

Corresponding parameters of our test environment are listed in Table 1.

**Table 1.** Parameters of corresponding scenarios.

| Parameter | Detail Information |
|---|---|
| Length of the loop line | 9 km |
| Number of 5G-R base stations | 5 |
| Height of BS antenna from rail track surface | 26 m |
| Gain of the BS antenna | 17.5 dBi |
| Gain of the on-board antenna | 0 dBi |
| Height of on-board roof antenna from rail track surface | 4.2 m |
| Speed of the testing locomotive | 80 km/h |
| Frequency band of 5G-R system | 1965–1975 MHz/2155–2165 MHz |
| Azimuth of K1.1 BS antenna (pointing to K4.2 direction) | 100° |
| Elevation angle of K1.1 BS antenna (pointing to K4.2 direction) | 5° |
| Azimuth of K4.2 BS antenna (pointing to K1.1 direction) | 340° |
| Elevation angle of K4.2 BS antenna (pointing to K1.1 direction) | 2° |

## 3. Channel Modeling Equipment and Data Processing Method

*3.1. In-Service SS-RSRP Signal-Based Passive 2100 MHz Large-Scale Channel Modeling*

3.1.1. Hardware of the Testing System

The hardware of the in-service SS-RSRP signal-based passive large-scale channel testing system consists of PC for data collection and processing, GNSS antenna for positioning and the 5G-R test scanner (R&S TSME6) connected to the 5G-R antenna, as shown in Figure 10. Of these, the 5G-R test scanner (R&S TSME6) is the key component, which can realize the real-time acquisition of the SS-RSRP parameters of the 5G-R network.

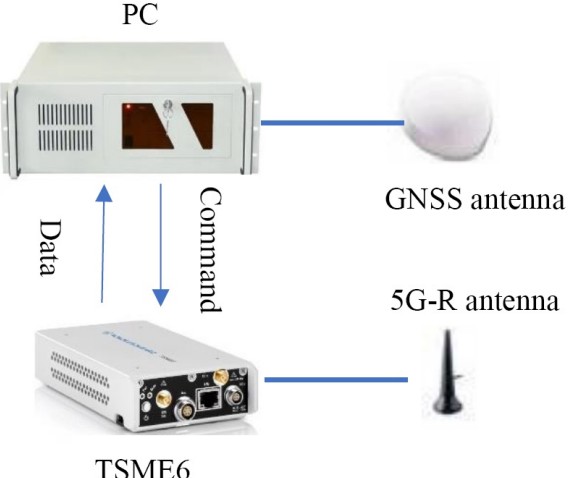

**Figure 10.** Hardware of in-service SS-RSRP signal-based passive large-scale channel testing system.

R&S TSME6 is a high-performance drive test scanner device produced by the ROHDE& SCHWARZ company, which is widely used for the channel measurement of wireless communication systems. The main parameters of R&S TSME6 are listed in Table 2.

**Table 2.** Parameters of TSME6.

| Parameters | Detail Information |
|---|---|
| Level measurement uncertainty | <1 dB |
| RF receive paths | 1 |
| SSB subcarrier spacings supported [1] | 15 kHz, 30 kHz |
| Sample rate | 20 samples per second |

[1] Sub-carrier spacing of the 5G-R system in our test campaign is 15 kHz.

3.1.2. Introduction of Large-Scale Channel Models

The FI (frequency-independent), CI (close-in), ABG (alpha beta gamma) and TR38.901 models are classic large-scale propagation channel models, and different models have different applicable scenarios and characteristics. Among them, the ABG model is an extension of the FI model for multiple frequencies, and under a single frequency condition, the ABG model is the same as the FI model, and since the 2100 MHz 5G-R network is a single frequency networking system, this study does not consider the ABG model. The other three models are analyzed below.

FI Model

This model assumes that path loss is frequency-independent, meaning that path loss only depends on distance and other environmental factors. The FI model path loss model has two parameters and does not consider the setting of reference points. The path loss formula of this model is as follows:

$$PL^{FI}(d)[\text{dB}] = \alpha + 10\beta \log_{10}(d) + X_\sigma^{\text{FI}}, \tag{1}$$

where $\alpha$ is the floating intercept in dB, and $\beta$ is the fitting slope, and $X_\sigma^{\text{FI}}$ is shadow fading that follows zero mean Gaussian distribution.

CI Model

The path loss of the CI model is proportional to the logarithm of the propagation distance. The CI model is commonly used in close-range communication scenarios, such as indoor communication systems [33]. The path loss formula of the CI model is as follows:

$$PL^{\text{CI}}(f,d)[\text{dB}] = FSPL(f,d_0) + 10\alpha \log_{10}\left(\frac{d}{d_0}\right) + X_\sigma^{CI}, \tag{2}$$

where $d_0 = 1$ m and $d \geq d_0$. $X_\sigma^{CI}$ is a zero mean Gaussian random variable with a standard deviation of $\delta$. The parameter $X_\sigma^{CI}$ represents the large-scale channel fluctuations caused by shadow effects. $f$ is the frequency in GHz, $d$ is the distance between the transmitting and receiving ends. In the CI model, the parameter $d_0$ is introduced based on physical distance.

$$FSPL(f,d_0) = 10 \log_{10}\left(4\pi d_0/\lambda\right)^2. \tag{3}$$

The $FSPL(f,d_0)$ value is only related to frequency. The parameter $\alpha$ is the road loss index (PLE). The CI path loss model can be used to estimate path loss based on cross-polarization or co-polarization measurements.

3GPP TR 38.901 Model

The 3GPP TR 38.901 model provides a detailed description of various wireless channel models used to simulate the wireless channel characteristics of 5G networks. These models include various environments and conditions, such as urban micro-cells (UMi), urban macro-cells (UMa), rural macro-cells (RMa), indoor hot-spots (InH), and so on. Due to the characteristics of a fast moving speed, less obstruction, small distance between base stations, a complex and variable environment (such as viaducts, cuttings, tunnels, etc.) and strip-shaped coverage along the railway, the RMa model meets the requirements of mainline railway scenarios based on the above analysis, and the received signal normally has a strong direct component. In this paper, the experimental environment is set up in an area without obvious obstructions to ensure a good line-of-sight transmission distance between the transmitting and receiving ends. The LOS model is selected for channel modeling, and the formula is as follows:

$$PL_1 = 20\log(40\pi d_{3D} f_c/3) + \min\left(0.03h^{1.72}, 10\right)\lg(d_{3D}) - \min\left(0.044h^{1.72}, 14.77\right) + \\ 0.002\lg(h)d_{3D} + X_\sigma^{TR:38.901}, \tag{4}$$

where $d_{3D}$ is the 3D distance, $f_c$ represents frequency, $h$ is the average height of buildings and $X_\sigma^{TR:38.901}$ is the Gaussian random variable.

Shadow Fading Formula

$X_\sigma$ conforms to a zero mean Gaussian distribution with standard deviation $\sigma_i$, $X_\sigma \sim N\left(0,\sigma^2\right)$, $\sigma_i$ is calculated as:

$$\sigma_i = \sqrt{\overline{X}^2}, \tag{5}$$

$$X = P_L(d) - \overline{P}_L(d), \tag{6}$$

where $P_L(d)$ and $\overline{P}_L(d)$ represent large-scale fading and path loss at the distance of $d$, respectively.

### 3.1.3. Model Parameter Fitting Algorithm

The SS-RSRP of base station K4.2, which represents the scenario of RRU mounted on the tower, is collected as the testing locomotive runs from K4.2 towards K1.1 on the test line of loop railway, and five sets of measurement data are collected during the testing as the locomotive runs for five laps, three sets of which are used as the training sets, whilst the other two sets are used as the verification sets. By adopting the least squares method, the training datasets are used to optimize the three formulas of the corresponding models of FI, CI and TR38.901, and the objective functions for each formula is constructed. By using the gradient descent algorithm, the corresponding objective functions serves to optimize the chances of obtaining optimal formula parameters. By using the verification datasets, the *RMSE* and *MAE* of the three formulas are calculated which serve to make the comparison of the fitting effects of different models on mainline railway 5G-R scenarios. The fitting process is shown as Figure 11.

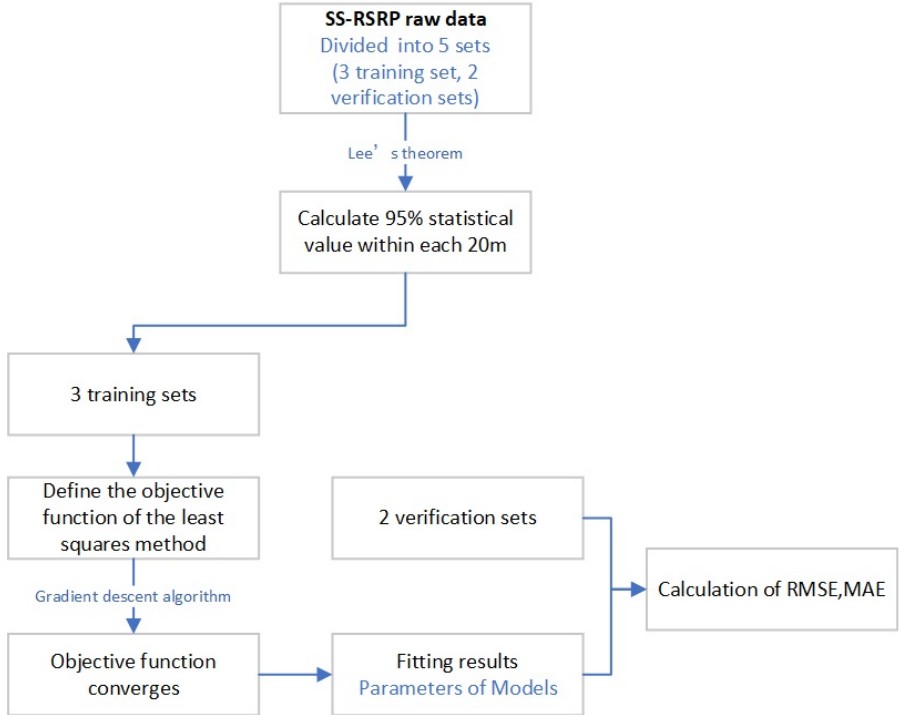

**Figure 11.** Fitting process of the channel parameters.

Data Collection and Processing

The SS-RSRP raw data are shown in panel *a* of Figure 12. The horizontal axis represents the distance from the base station K4.2, and the vertical axis represents the power in dBm. After a distance of 1.8 km, the attenuation of received signal power is significant, because there is EMU parking garage, which is a monomer construction with a height of 11 m, and a span of 500 m wide, as shown in the red circle of Figure 12, which blocks the LOS signal. In order to achieve better fitting results, we excluded data from 1.8 km onwards.

The received signal level is composed of a combination of small-scale fading signals and large-scale fading signals. Therefore, the statistical interval cannot be selected too small, otherwise small-scale fading components cannot be filtered out. According to Lee's theorem [34], in order to effectively achieve the goal of "eliminating fast fading and preserving slow fading", the influence of fast fading components on signal statistics should not exceed 1dB, and the statistical interval should be at least 40 $\lambda$. The carrier frequency of our test is $f$ = 2160 MHz, and the statistical interval should be greater than 6 m. In the meanwhile, the statistical interval should not be too large, which results in the loss of statistical information. Taking into account the above factors, the statistical interval is set at

20 m, which means that the 95% value within the interval is taken every 20 m. This not only ensures that the impact of fast fading components on signal statistical values is not greater than 1 dB, but also facilitates characterizing the distribution of the wireless field strength coverage along the railway line.

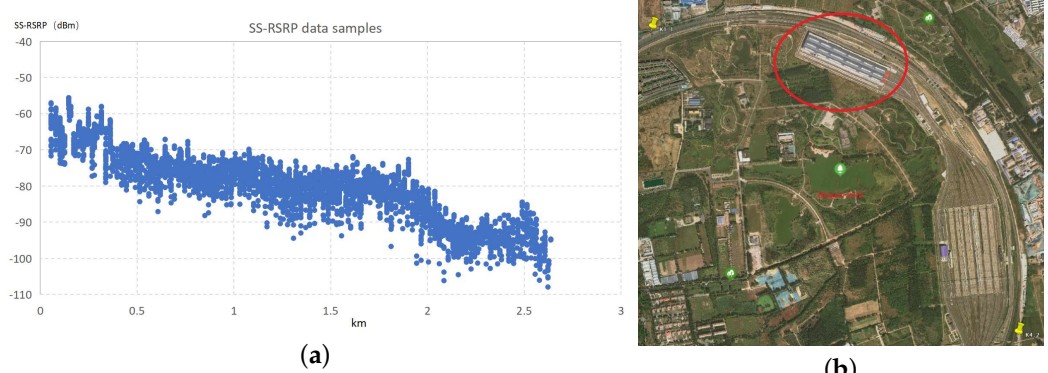

(**a**)

(**b**)

**Figure 12.** Raw SS–RSRP data of K4.2: (**a**) Raw SS–RSRP data of K4.2; (**b**) Building at the position of K1.98~K2.48.

Fitting Process

The least squares method is adopted for formula fitting, and the objective function defined in the fitting process is:

$$obj = \frac{1}{n}\sum_{i=1}^{n}(y_{true.i} - y_{pred,i})^2 \quad , \tag{7}$$

where $y_{true}$ represents the actual measured value. And, the predictive value of the three models are:

$$y_{pred,FI} = \alpha + 10\beta \log_{10}(x) + X_{\sigma}^{FI}, \tag{8}$$

$$y_{\text{pred, }CI} = 10\log_{10}(\frac{4\pi d_0}{\lambda})^2 + 10\alpha \log_{10}(\frac{d}{d_0}) + X_{\sigma}^{CI}, \tag{9}$$

$$y_{pred,TR38.901} = 20\log_{10}(40\pi x f_c/3) + \alpha h^{\beta}\log_{10}(x) - \delta h^{\beta} + \gamma \log_{10}(h)x + X_{\sigma}^{TR38.901}. \tag{10}$$

The initial values of $\alpha$, $\beta$, $\sigma$, $\gamma$ are all set to 1, and then their derivatives and corresponding iterative formulas are calculated, respectively:

$$\alpha_i = \alpha_{i-1} - (\text{lr} * \text{d}\alpha_{i-1}), \tag{11}$$

$$\beta_i = \beta_{i-1} - (\text{lr} * \text{d}\beta_{i-1}), \tag{12}$$

$$\delta_i = \delta_{i-1} - (\text{lr} * \text{d}\delta_{i-1}), \tag{13}$$

$$\gamma_i = \gamma_{i-1} - (\text{lr} * \text{d}\gamma_{i-1}), \tag{14}$$

where $l_r$ represents the learning rate which is set to 0.01, whilst the number of iterations is set to 500 and the algorithm will complete when it converges.

Evaluation Indicators for Fitting Results

*RMSE* and *MAE*, which are two typical regression evaluation indicators, are adopted as evaluation indicators for evaluating the fitting results. *RMSE* represents the difference between the regression results and the actual measured values, with the larger the error, the greater the value; As for *MAE*, the value 0 represents a perfect model, and the smaller the MAE value, the better the fitting effect.

$$RMSE = \sqrt{\frac{1}{n}\sum_{i=1}^{n}(\hat{y}i - yi)^2}, \tag{15}$$

$$MAE = \frac{1}{n}\sum_{i=1}^{n}|\hat{y}_i - y_i|. \tag{16}$$

*3.2. SDR-Based 2100 MHz 5G-R Channel Modeling*

3.2.1. Hardware of the System

The hardware of the SDR-based 2100 MHz 5G-R channel modeling system consists of transmitting and receiving ends. The transmitting end is constitute by a PC for data collection and processing, a USRP-2954 which is connected to the 5G-R antenna as the transmitter, and a power amplifier (PA). As for the receiving end, the equipment are almost the same, except that the PA is replaced by a low-noise amplifier (LNA), as shown in Figure 13.

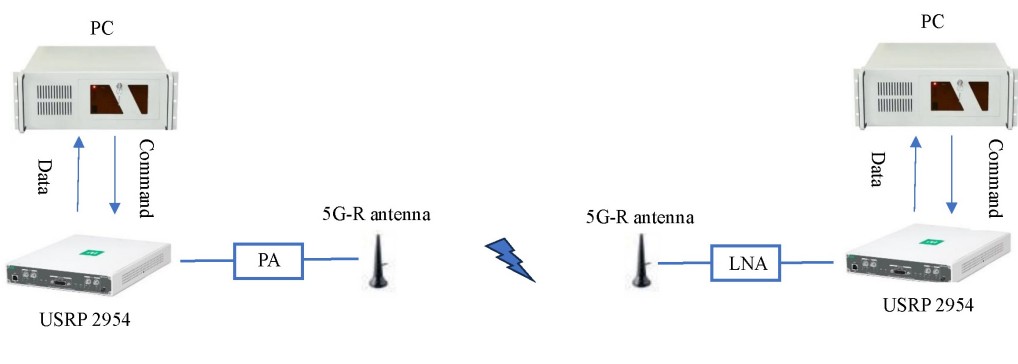

**Figure 13.** Hardware of the SDR-based 2100 MHz 5G-R channel modeling system.

The USRP-2954 is the key component of both ends, which can realize the real-time transmission and reception of the ZC sequence at the transmitting and receiving end, respectively.

The key parameters of USRP-2954 are listed in Table 3:

The output power of USRP-2954 is 20 dBm, and we use a 26 dB power amplifier to increase the output power of the transmitter to 46 dBm (40 W), which is equal to the output power of each channel of the 2100 MHz 5G-R 8T8R RRU equipment.

**Table 3.** Parameters of USRP-2954.

| | Parameter | Detail Information |
|---|---|---|
| | Frequency Range | 10 MHz–6 GHz |
| | Gain range | 0 dB–31.5 dB |
| Transmitter | Maximum instantaneous real-time bandwidth | 160 MHz |
| | Maximum I/Q sample rate | 200 MS/s |
| | Digital-to-analog converter (DAC) resolution | 16 bit |
| | Output power | 20 dBm (100 mW) |

**Table 3.** *Cont.*

| | Parameter | Detail Information |
|---|---|---|
| | Frequency range | 10 MHz–6 GHz |
| | Gain range | 0 dB–37.5 dB |
| | Maximum input power | −15 dBm |
| Receiver | Noise figure | 5 dB–7 dB |
| | Maximum instantaneous real-time bandwidth | 160 MHz |
| | Maximum I/Q sample rate | 200 MS/s |
| | Analog-to-digital converter (ADC) resolution | 14 bit |

### 3.2.2. Data Processing Method

The corresponding algorithms, ZC sequence selection as well as the discrete algorithm of cross-correlation, are the same as those in [35], which was recently published by our team. The ZC sequence is sent from the transmitting end, and at the receiving end, the received ZC sequence is used to perform a discrete cross-correlation operation with local ZC to obtain the result of CIR. The channel impulse response (CIR) process flow is shown as Figure 14:

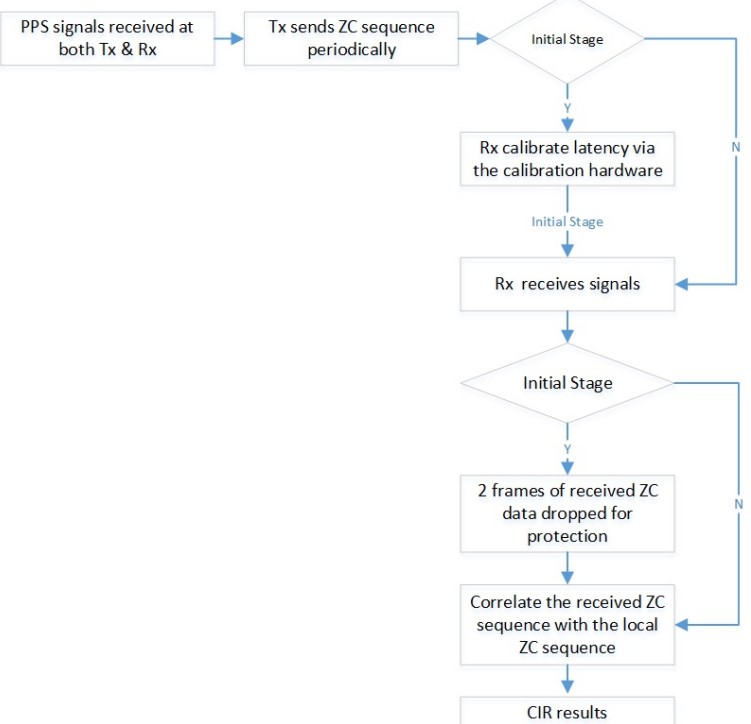

**Figure 14.** Channel impulse response (CIR) process flow.

Based on the platform of LabVIEW (2021 SP1), the software used for channel parameter deriving is developed, which is shown as Figure 15:

In Figure 15, the horizontal axis of the picture at the upper right is delay, the vertical axis is power, and the green curve indicates the received signal power at different delay positions.The picture in the middle of the right side is a waterfall map, with time delay on the horizontal axis and time flow on the vertical. Different colors represent the power intensity at different times, and the warmer the tone, the higher the power. The picture at the bottom right is a fluorescence picture, with time delay on the horizontal axis and power on the vertical axis. The warmer the color tone, the greater the probability of the power intensity of the time delay.

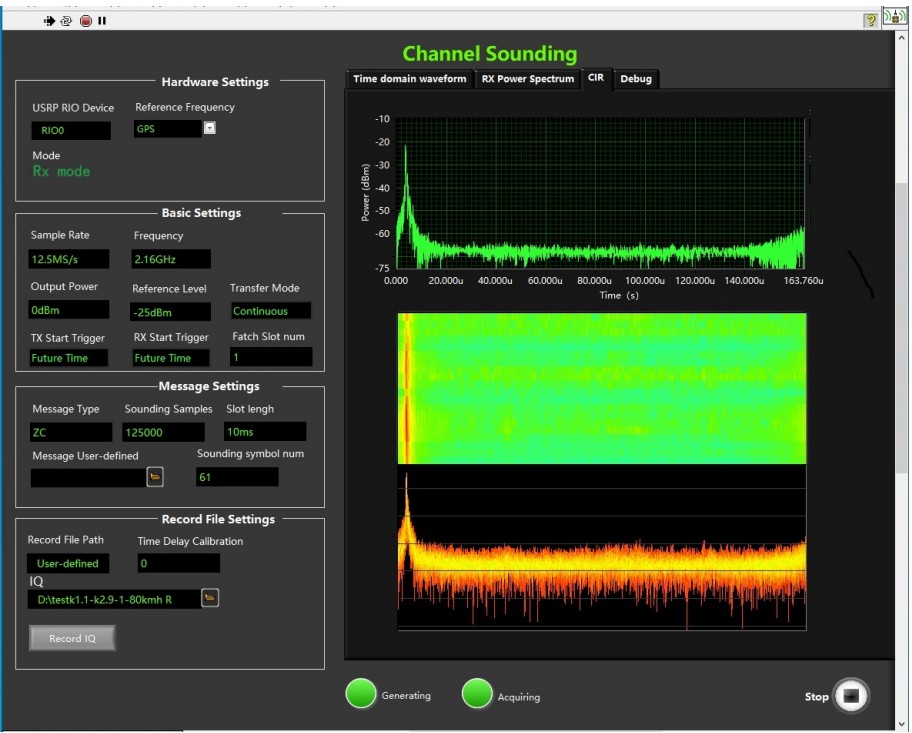

**Figure 15.** Self–developed software for SDR channel sounding.

The time domain waveform of the ZC sequence at the transmitting and receiving ends is shown as Figure 16.

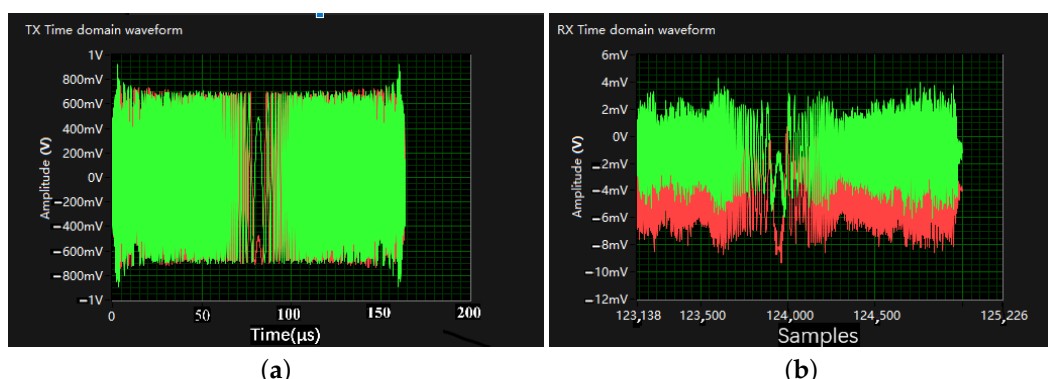

(**a**)                                    (**b**)

**Figure 16.** Time domain waveform of ZC sequence: (**a**) transmitting end; and (**b**) receiving end.

The two colors of curves in the figure represent In-Phase and Quadrature signal respectively.

Based on the mentioned SDR-based 2100 MHz 5G-R channel modeling system and corresponding data processing methods, the testing task was carried out with the cooperation of on-board and trackside testing staff, as shown in Figure 17. Here, the antenna and feeder of K1.1 is disconnected for the base station RRU equipment, and connected to our test system, to represent the scenario of RRU mounted under the tower.

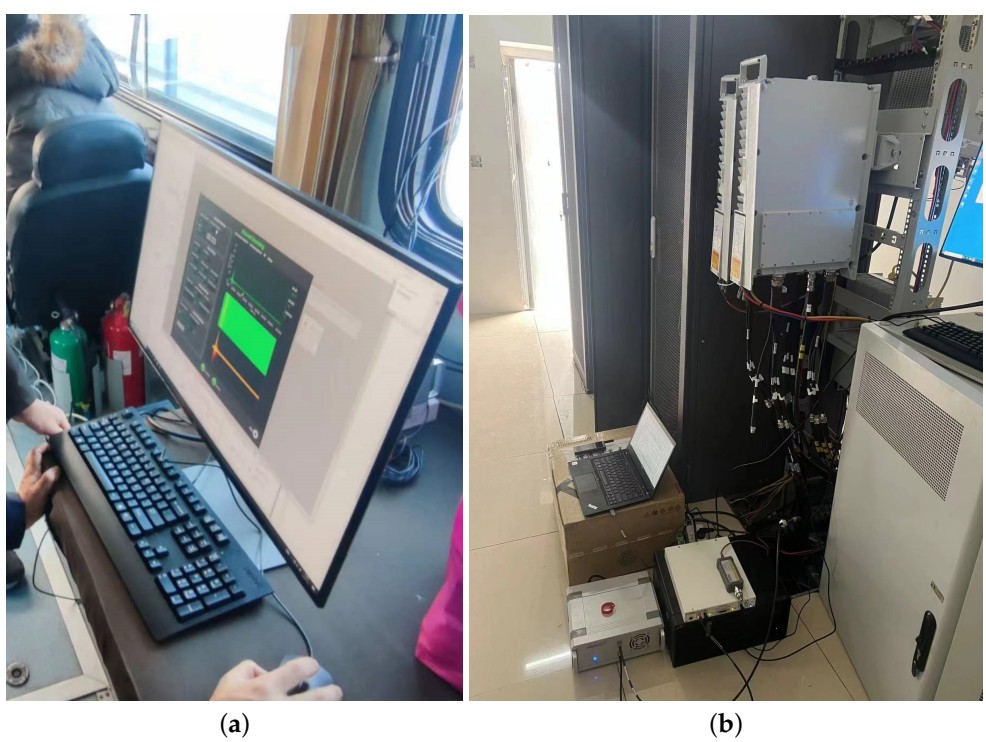

**Figure 17.** On–board and trackside testing equipment. (**a**) On–board testing equipment. (**b**) Trackside testing equipment in the equipment room of BS K1.1.

## 4. Results

Based on the derived channel parameters, further analyses are conducted with respect to two aspects:

### 4.1. Fitting Results and Comparison of FI, CI and TR 38.901 Models Based on Passive Measurement

#### 4.1.1. Fitting Result of FI Model

When $\alpha = 29.162$ and $\beta = 1.897$, the highest fitting degree is achieved, and the FI model formula is:

$$PL^{FI}(d)[\text{dB}] = 29.162 + 18.97 \log_{10}(d) + X_\sigma^{\text{FI}}. \tag{17}$$

And, $RMSE = 3.945$, $MAE = 3.027$ and the shadow fading standard deviation $\sigma_{FI} = 3.279$.

#### 4.1.2. Fitting Result of the CI Model

When $\alpha = 1.329$, the highest fitting degree is achieved, and the CI model formula is:

$$PL^{\text{CI}}(f,d)[\text{dB}] = 10 \log_{10}(4\pi d_0/\lambda)^2 + 13.29 \log_{10}\left(\frac{d}{d_0}\right) + X_\sigma^{\text{CI}}. \tag{18}$$

And, $RMSE = 4.073$, $MAE = 3.037$ and the shadow fading standard deviation $\sigma_{CI} = 3.398$.

#### 4.1.3. Fitting Result of TR 38.901 Model

When $\alpha = 0.179$, $\beta = 0.367$, $\gamma = 0.807$, $\sigma = 9.367 \times 10^{-6}$, the highest fitting degree is achieved, and the TR 39.901 model formula is:

$$PL_1 = 20\log(40\pi d_{\mathfrak{g}D}f_c/\mathfrak{g}) + \min\left(0.0999h^{0.851}, 10\right)\lg(d_{\mathfrak{S}D}) -$$
$$\min\left(1.007h^{0.851}, 14.77\right) + 0.905\lg(h)d_{\mathfrak{S}D} + X'^{\mathrm{TR38.901}}_\sigma. \tag{19}$$

And, $RMSE = 3.857$, $MAE = 3.017$ and the shadow fading standard deviation $\sigma_{TR38.901} = 3.311$.

### 4.1.4. Comparison of Models

The $RMSE$ and $MAE$ values of FI, CI and TR 38.901 models are shown in Table 4. The $RSME$ and $MAE$ values of the three models are all fairly low, indicating that the fitting effect of the models has achieved satisfactory results.

**Table 4.** $RMSE$ and $MAE$ of FI, CI and TR 38.901 models.

| Indicator<br>Models | RMSE | MAE |
| --- | --- | --- |
| FI | 3.945 | 3.027 |
| CI | 4.073 | 3.037 |
| TR 38.901 | 3.857 | 3.017 |

As shown in Figure 18, the output values of the fitted FI, CI and TR 38.901 formulas are represented by green, blue and orange curves, respectively. And, the 95% statistical values of the actual measured data, which are divided into training sets and verification sets, are represented by dots of different colors in the Figure.

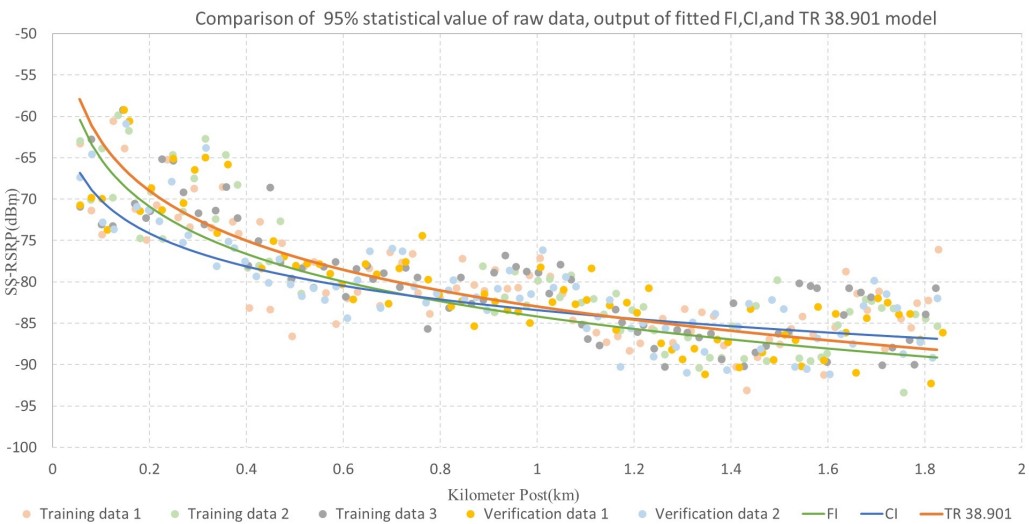

**Figure 18.** Comparison of the three fitted models.

Based on the above analysis, it can be concluded that the TR 38.901 model, which is specifically customized for the characteristics and requirements of 5G systems, has the lowest $RMSE$ and $MAE$ values, and the model can best adapt to the mainline railway 5G-R scenario.

### 4.2. Channel Characteristics Derived from SDR-Based Channel Modeling

Using the hardware and data processing method mentioned in Section 3.2, 60 groups of ZC sequences with the length of 2048 points within 10ms are used for PDP calculation, which means that 60 samples are collected within each 10 ms time span, and based on each sample, the received power, the latency of each path are calculated.

We find the algorithm which can effectively identify an effective path, in which a power margin is defined as:

$$P_{margin} = \frac{\sum_{m=n+1}^{n+100} P_m}{100} + 0.7 \times Max(P_{n+1}, P_{n+2}......, P_{n+100}), \tag{20}$$

where $Power_{margin}$ is the defined power margin if the received power of a path is greater than the margin, and if it is greater than $-90$ dBm, it is defined as an effective path.

As shown in Table 5, the parameters derived for one out of 60 samples within 10 ms span are listed, in which we can see that, at the position of K1.1, eight paths are detected, and the main path (LOS) has a received power of $-20.71$ dBm and latency of 147 ns, and the weakest path has a received power of $-52.051$ dBm and latency of 3587 ns.

**Table 5.** Parameters derived from one sample out of 60 samples within 10 ms at K1.1.

| Latency (ns) | Received Power (dBm) | Longitude | Latitude | Timestamp (s) | Speed (km/h) | Position (km) |
|---|---|---|---|---|---|---|
| 147 | $-20.71$ | 116.519385 | 40.0017217 | 142,417.01 | 80.228 | 1.1 |
| 627 | $-33.283$ | 116.519385 | 40.0017217 | 142,417.01 | 80.228 | 1.1 |
| 1187 | $-35.47$ | 116.519385 | 40.0017217 | 142,417.01 | 80.228 | 1.1 |
| 1427 | $-37.462$ | 116.519385 | 40.0017217 | 142,417.01 | 80.228 | 1.1 |
| 2147 | $-42.264$ | 116.519385 | 40.0017217 | 142,417.01 | 80.228 | 1.1 |
| 2307 | $-48.755$ | 116.519385 | 40.0017217 | 142,417.01 | 80.228 | 1.1 |
| 2467 | $-48.998$ | 116.519385 | 40.0017217 | 142,417.01 | 80.228 | 1.1 |
| 3587 | $-52.051$ | 116.519385 | 40.0017217 | 142,417.01 | 80.228 | 1.1 |

Based on these samples, the relationships between the received power and distance, MPCs and distance, as well as RMS delay spread and distance, are deeply analyzed below:

### 4.2.1. Relationship between Number of MPCs and Distance

The multi-path components (MPCs) at different positions are shown in Figure 19, in which we can tell, as the distance between transceivers increases, the number of MPCs decreases. Multi-path components are the richest within a distance range of 300 m, with a maximum of 19 MPCs, which appears at the position of both 53 m and 225 m, a minimum of 3 MPCs, and a 50% statistical value of 11 MPCs. When the distance is larger than 1049 m, the scene of only one main path (LOS) begins to appear, and the probability gradually increases as the distance increases.

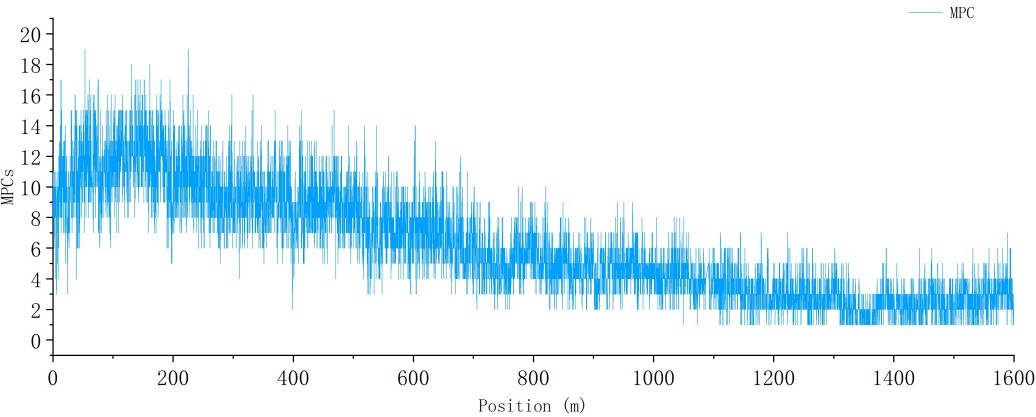

**Figure 19.** MPCs at different positions.

### 4.2.2. Relationship between Received Power and Distance

By adding the received power of all paths at each data collecting position, we obtain the total received power of different positions, which is shown in Figure 20:

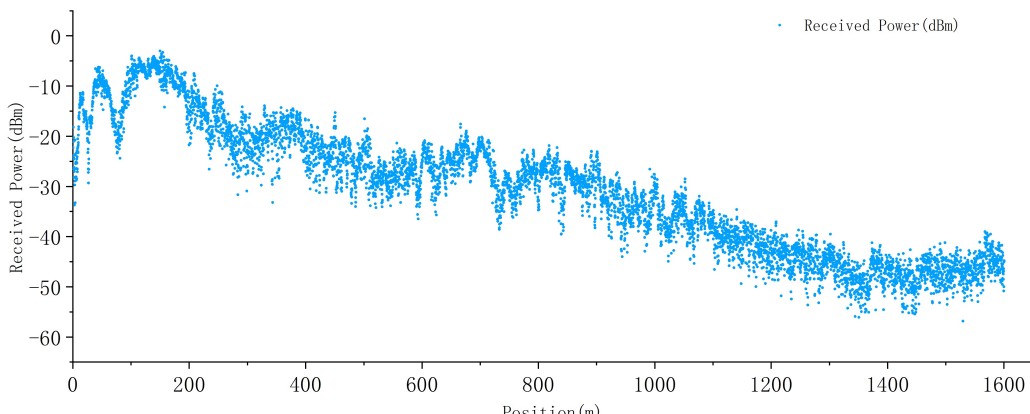

**Figure 20.** Raw data of the total received power.

In Figure 20, we can tell that, when the distance is larger than 1500 m, the received power increases with the increase in distance between the transceivers, which is because the test line is loop-shaped, and at the far end the propagation characteristic, no long conforms to the linear scenario; thus, in our subsequent analysis, the data after 1500 m are removed from our dataset for analysis.

By calculating the 95% statistical value of the total received power within each 20 m, we obtain Figure 21:

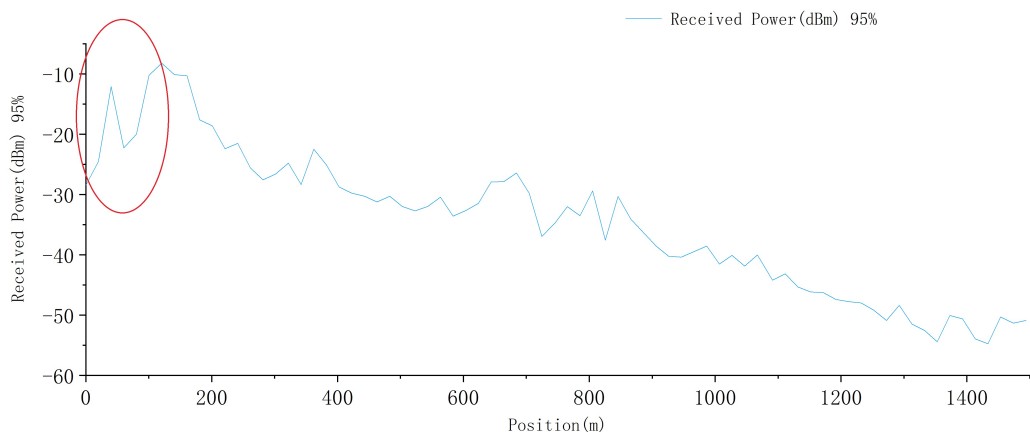

**Figure 21.** The 95% statistical value of total received power.

As analyzed in Section 4.2.1, at each sampling position, there is an LOS path, and several N-LOS paths, so under the specific mainline railway 5G-R 2100 MHz scenario, the received power propagation characteristic conforms to Rician distribution.

The Python Rician Fitting function is used to fit the Rician distribution with the 95% statistical value of total received power. Moreover, since the data in the near-field do not conform to Rician distribution, the data before 149 m were removed from our fitting data set, as seen in the red circle of Figure 21. The fitting result is shown in Figure 22.

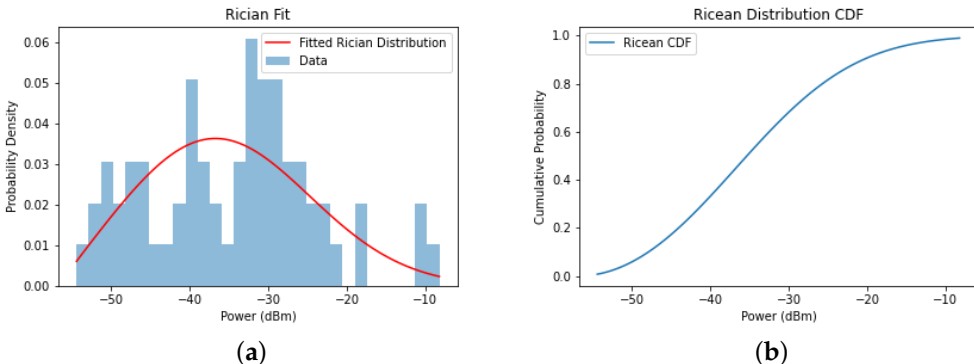

**Figure 22.** Rician distribution fitting results: (**a**) fitted Rician distribution; (**b**) CDF of Rician distribution.

The formula of the Rician probability density function and probability distribution function are as follows:

$$f(x; b, loc, scale) = \frac{x - loc}{scale^2} \cdot \exp\left(-\frac{(x - loc)^2 + b^2}{2 \cdot scale^2}\right) \cdot I_0\left(\frac{b \cdot (x - loc)}{scale^2}\right), \tag{21}$$

$$F(x; b, loc, scale) = 1 - \exp\left(-\frac{(x - loc)^2 + b^2}{2 \cdot scale^2}\right) \cdot I_0\left(\frac{b \cdot (x - loc)}{scale^2}\right), \tag{22}$$

where $I_0$ is a zero-order Bessel functions, $b$ is the shape parameter which is related to the shape of the curve, $loc$ is the location parameter which determines position of the curve on the horizontal axis and $scale$ is the scale parameter which is related to the vertical amplitude. And, according to our fitting result, the values of these three parameters are: $b = 1.341659834276076; loc = -56.81054171751569; scale = 12.600749936636792$.

The Rician channel K-Factor is defined as the logarithm of the power ratio between the LOS path and sum of reflected multiple paths:

$$K = 10 \times \lg\left(\frac{P_{LOS}}{\sum_{n=1}^{m} P_n}\right), \tag{23}$$

where $P_{LOS}$ is the power of the LOS path in W, m is the number of N-LOS paths and $P_n$ is the power of the nth N-LOS path in W. The larger the K-factor, the smaller the impact of the multi-path on the LOS path during transmission. K-factor is an important indicator that can be used to evaluate the system performance and optimize the system design.

The K-factors in dB at different positions are shown in Figure 23:

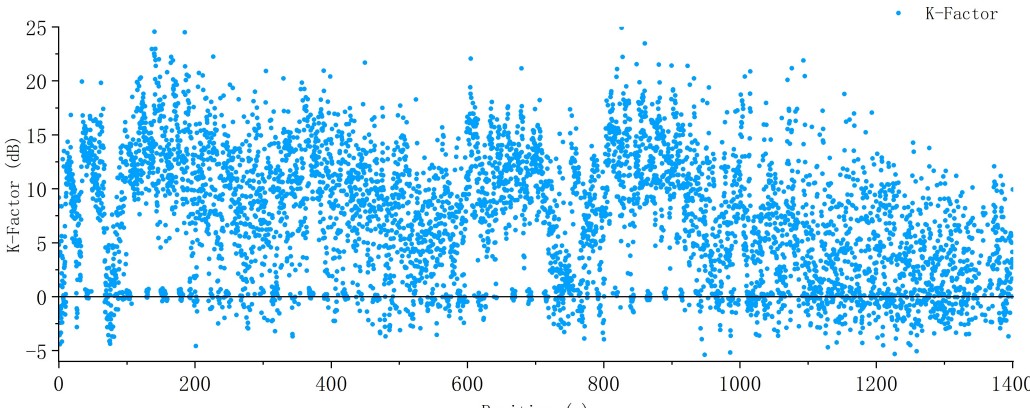

**Figure 23.** K-factors at different positions.

In Figure 23, we can tell that K-factor does not change obviously with distance. The maximum value of 24.924 dB appears at the position of 826 m, and the minimum value of −5.389 dB appears at the position of 948 m. The 50% statistical value is 7.696 dB.

### 4.2.3. Relationship between RMS Delay Spread and Distance

Root mean square delay spread (RMS delay spread) reflects the richness of the surrounding scatterers in the propagation environment, and is defined as:

$$RMS_{DS} = \sqrt{\frac{1}{P_i} \cdot \sum_{l=1}^{L} P_l(\tau_l)^2 - (\frac{1}{P_i} \cdot \sum_{l=1}^{L} P_l \cdot \tau_l)^2}, \tag{24}$$

where $P_i$ is the total received power, $L$ is the number of total paths, $P_l$ and $\tau_l$ are the received power and latency of the lth path, respectively.

The values of the RMS delay spread in ns at different positions are shown in Figure 24:

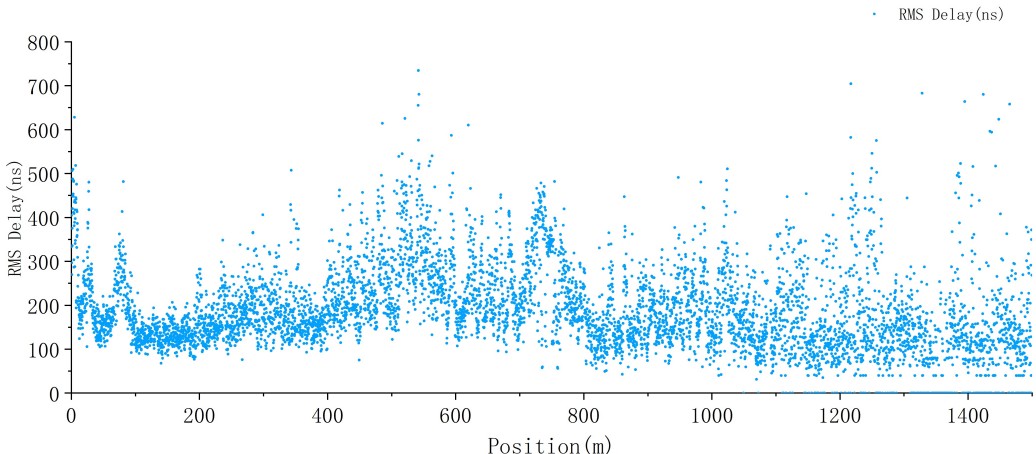

**Figure 24.** RMS delay spread at different positions.

In Figure 24, we can tell that RMS delay spread does not change obviously with distance. The maximum value of 735 ns appears at the position of 541 m, and the minimum value of 31 ns appears at the position of 1069 m. The 50% statistical value is 173 ns.

The CDF and PDF of RMS delay spread are shown in Figures 25 and 26:

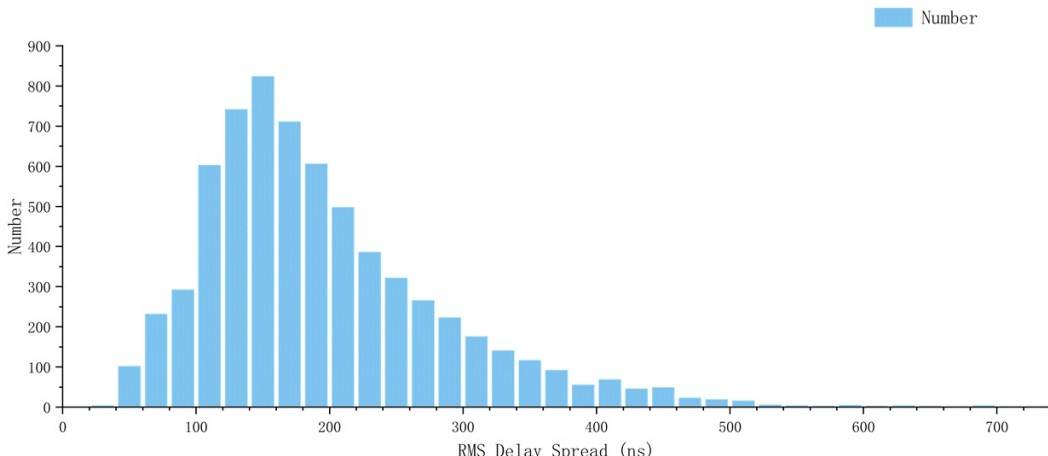

**Figure 25.** PDF of RMS delay spread.

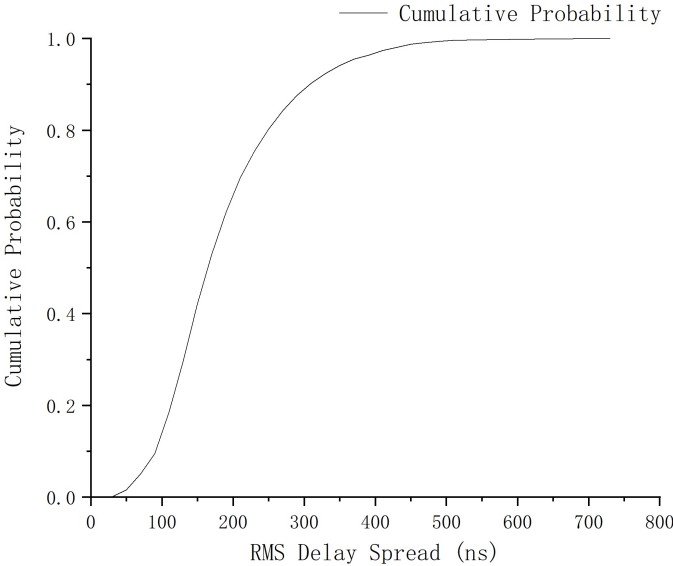

**Figure 26.** CDF of RMS delay spread.

## 5. Conclusions

Both the in-service SS-RSRP signal-based passive 2100 MHz large-scale channel testing system and SDR-based 2100 MHz 5G-R channel sounding system are adopted in our measurement campaign conducted in the 5G-R dedicated network testing environment along the test line of the loop railway of the National Railway Track Test Center of China, as well as the 2100 MHz 5G-R mainline railway large- and small-scale channel characteristics are derived and deeply analyzed.

(1) With data derived from the passive testing system, the FI, CI as well as TR 38.901 large-scale channel models are properly fitted, and low $RMSE$, $MAE$ as well as shadow fading standard deviation $\sigma$ indicates that the fitting effects of all three models achieved satisfactory results, among which, the fitted TR 38.901 model has the lowest $RMSE = 3.857$ and $MAE = 3.017$ values, and its shadow fading standard deviation $\sigma_{TR38.901} = 3.311$, which means that the TR 38.901 model can best adapt to the railway 5G-R scenario.

(2) Based on the SDR-based channel sounding system, conclusions on the relationship between the parameters of MPCs, received power, RMS time spread and the distance between transceivers are drawn. As the distance between transceivers increases, the number of MPCs decreases, and multi-path components are the richest within a 300 m range of distance, with a maximum of 19 MPCs and a 50% statistical value of 11 MPCs. The 95% statistical value of the total received power conforms with Rician distribution, and the PDF and CDF of the Rician distribution are given. The K-factor is calculated, for which the maximum value is 24.92 dB and the minimum value is −5.389 dB, whilst its 50% statistical value is 7.696 dB. Moreover, the RMS delay spread does not change obviously with distance, the relationship between RMS delay spread and distance, the maximum value of 735 ns, the minimum value of 31 ns as well as the 50% statistical value of 173 ns, whilst PDF and CDF of the RMS delay spread are given.

The conclusions of this article are helpful for R&D, deployment as well as the network optimization of 2100 MHz 5G-R-dedicated networks in the context of a railway mainline scenario. In the meanwhile, the relevant testing and analysis method employed in this letter can be extended to study other network standards, such as LTE, 6G, as well as other frequency bands, for instance, the 1900 MHz of FRMCS in Europe, and also can be used in studying of more railway scenarios. Moreover, four aspects of future work are pointed out:

(1) The SDR-based channel modeling in this letter is actually SISO channel modeling, and the angle information of the 2100 MHz 5G-R channel characteristics is not included, which can be further studied.

(2) The speed of our testing locomotive running on the loop railway line is 80 km/h, which is far below the operating speed of HSR. The relevant testing systems as well as methods can be further validated on high-speed trial lines under more diverse scenarios at 350 km/h speed, and more realistic models can be achieved.

(3) Antenna technology is an important factor affecting the propagation of radio waves. In this letter, a 2100 MHz 5G-R plate antenna was used in the railway mainline scenario. At present, due to its good directionality and high antenna efficiency, public network operators have conducted pilot application research on lens antennas. In the future, research on the application of antenna technologies such as lens antennas can be carried out for typical railway scenarios to improve channel availability at the equipment level. In the meanwhile, the roof of the testing locomotive is metal, which is a large metal reflector that affects the directional pattern of the roof antenna, and the numerical impact can be further studied in the future.

(4) The established 2100 MHz 5G-R channel model can be imported into channel simulation platforms, such as Keysight PROPSIM series products, to assist in 2100 MHz 5G-R BS and UE equipment R&D, as well as inspection and verification.

**Author Contributions:** Conceptualization, Y.L. (Yiqun Liang) and H.L.; methodology, Y.L. (Yiqun Liang), A.L. and Y.L. (Yi Li); software, Y.L. (Yiqun Liang); validation, Y.L. (Yiqun Liang) and H.L.; formal analysis, Y.L. (Yiqun Liang) and H.L.; writing—original draft preparation, Y.L. (Yiqun Liang); writing—review and editing, Y.L. (Yiqun Liang); visualization, Y.L. (Yiqun Liang); supervision, Y.L. (Yiqun Liang) and H.L.; project administration, Y.L. (Yiqun Liang) and H.L. All authors have read and agreed to the published version of the manuscript.

**Funding:** This research was funded by the Science and Technology R&D Program of China State Railway Group Co., Ltd., grant number SY2021G001, the Science and Technology R&D Program of China Academy of Railway Sciences Co., Ltd., grant number of 2022YJ033, 2023YJ074.

**Data Availability Statement:** Data are contained within the article.

**Acknowledgments:** This research was supported by Liu Liang from Beijing Zengyi Technology Company in terms of the validation of the SDR measurement system.

**Conflicts of Interest:** Authors Liang Yiqun, Li Hui, Li Yi, Li Anning were employed by the company China Academy of Railway Sciences Corporation Limited. The authors as well as Liu Liang from Beijing Zengyi Technology Company declare that the research was conducted in the absence of any commerial or financial relationships that could be construed as a potential conflict of interest.

## Abbreviations

The following abbreviations are used in this manuscript:

| | |
|---|---|
| 3GPP | The 3rd Generation Partnership Project |
| 5G-R | 5G for Railway |
| ABG | Alpha Beta Gamma |
| ADC | Analog-to-Digital Converter |
| BS | Base Station |
| CI | Close In |
| CIR | Channel Impulse Response |
| CSI | Channel State Information |
| DAC | Digital-to-Analog Converter |
| DL | Deep Learning |
| EMU | Electric Multiple Unit |
| FDD | Frequency Division Duplexing |
| FI | Frequency Independent |
| FRMCS | Future Railway Mobile Communication System |
| GBSM | Geometry-Based Stochastic Model |
| GPSDO | GPS Disciplined Oscillator |
| GSMA | Global System for Mobile communications Association |
| GSM-R | GSM for Railway |

| | |
|---|---|
| HSR | High-Speed Railway |
| I/Q | In-Phase/Quadrature |
| ITU | International Telecommunication Union |
| LabVIEW | Laboratory Virtual Instrument Engineering Workbench |
| LNA | Low-Noise Amplifier |
| LOS | Line of Sight |
| MAE | Root Mean Square Error |
| MIIT | Ministry of Industry and Information Technology (China) |
| MPC | Multi-Path Component |
| PA | Power Amplifier |
| PC | Personal Computer |
| PDP | Power Delay Profile |
| PL | Path Loss |
| QoS | Quality of Service |
| RIS | Reconfigurable Intelligent Surface |
| RMS | Root Mean Square |
| RMSE | Root Mean Square Error |
| RRU | Remote Radio Unit |
| RT | Ray Tracing |
| SDR | Software-Defined Radio |
| SISO | Single-Input–Single-Output |
| SNR | Signal-to-Noise Ratio |
| SS-RSRP | Synchronization Signal Reference Signal Received Power |
| T2T | Train-to-Train |
| TDL | Tapped Delay Line |
| UE | User Equipment |
| UIC | International Union of Railway |
| ZC Sequence | Zadoff–Chu Sequence |

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
