# Peer review of "Mainline Railway Modeled with 2100 MHz 5G-R Channel Based on Measured Data of Test Line of Loop Railway"

_symmetry, doi:10.3390/sym16040431_

Round 1

Reviewer 1 Report

Comments and Suggestions for Authors

The paper investigates the implementation of 5G for railway communication, specifically in the 2100MHz frequency band, which is globally recognized for 5G-R deployment. Conducted along the loop railway of the National Railway Track Test Center of China, the study employs both passive large-scale channel testing and a Software-Defined Radio (SDR)-based channel sounding system. The analysis includes deriving and deeply analyzing large-scale and small-scale channel characteristics, fitting classical channel models, and drawing conclusions about the relationship between various parameters using SDR-based measurements. The research aims to facilitate the design, deployment, and optimization of future mobile communication systems in railway mainline scenarios.

The paper is well written and well organized. However, there are some points that should be addressed.

1.      Some grammar errors are identified please correct.

2.      Commas and dotes after equation should be placed.

3.      I would not recommend having pictures of the authors please provide just the experiment results. Figure 15 should be modified.  

4.      All the Where after equations should be where with small letter. Noting that comma should be stated after the equation.

5.      The constitution should be further highlighted and provided as bullet points.

6.      The author needs to state the paper finding in the last part of introduction. For example what have been achieved in the results? Could be stated in subsection 1.2.

7.       The contribution is not clear in comparison to the state of the art methods in the introduction. I would suggest make it clear what have been done in comparison to the state of the art methods showing the contribution of the paper. Therefore, the novelty of the paper in comparison with the state of the art needs to be further highlighted. Just highlighted to be clarified to the reader.

8.      Could be please identify future works.

Comments on the Quality of English Language

We identify some grammar mistake and punctuation errors. Please modify and proofread the paper again. 

Author Response

Dear expert,

Thank you so much for taking the time to review this manuscript! Please see the attachment.

Reviewer 2 Report

Comments and Suggestions for Authors

Both an operational SS-RSRP signal-based passive 2100 MHz large-channel test system and an SDR-based 2100 MHz 5G-R channel probing system are used in the measurement campaign.The system is installed in a dedicated 5G-R network test bed along the railway line.The test line is carried out on the loop track and 2100 MHz 5G-R main line of China National Railway Track Testing Center, and both large and small channel characteristics are derived and analyzed in detail.

The paper is rigorous, well written and important. 

The authors are suggested to explore whether bio-inspired models could give a new orientation and dimension.

Author Response

Dear expert,

Thank you very much for taking the time to review this manuscript. And thank you again for your suggestion with respect to the bio-inspired models. I will carefully study the bio-inspired models and see how it works in the railway radio propagation scenario.

Reviewer 3 Report

Comments and Suggestions for Authors

The paper is devoted to the study of the conditions for the propagation of radio signals in railway traffic. The problem is relevant, relevant and classical in its formulation for radio communications. This is a necessary stage in the development of any radio communication system.

1. Main conceptual note: it is not clear why the article was submitted to the Symmetry journal? The first paragraph provides formal arguments about symmetry, but any ideal radio channel has this property, while a non-ideal one does not have it at all due to the asymmetry of the receiving and transmitting equipment. Are there really no specialized journals in MDPI on the topic of channel state measurement? Theses about the symmetry of the radio channel look artificial; another motivation is needed, at least correlating with the problems of symmetry at the level of physics.

2. The measurement procedure is unclear. The carriage was moving and what was being recorded? Did the base stations emit the same known signal? This is unclear from the text of the paper.

3. How was the antenna radiation pattern taken into account? Does the figure 8 show the radiation pattern of the antenna by itself, or the pattern when it is on the roof of the carriage? Is the roof of the carriage metal? 

4. Axes labels in data set figures are too small.

Author Response

Dear expert,

Thank you so much for taking the time to review this manuscript!

Round 2

Reviewer 1 Report

Comments and Suggestions for Authors

The author has addressed all the issues.  

Author Response

Dear Expert,

Thank you again for your time! I noticed that your latest comment is "The author has addressed all the issues.  ", so thers is no update of the former point-by-point response.

Best wishes!

Reviewer 3 Report

Comments and Suggestions for Authors

Dear authors, please upload the corrected version of your article. The PDF version of the article that I can download from my MDPI account contains a lot of notes in the PDF that makes the article very difficult to read. Could you please make all the corrections to the text in the usual style. As far as I know, according to MDPI rules, the corrected text is simply highlighted in color. It is assumed that the version that is provided to reviewers is the same version that is ultimately published.

Author Response

Dear Expert,

Thank you again for your time! Sorry for the previous uploaded PDF version, I have  I have re uploaded the PDF file, in which the changed parts corresponding to the review report is highlighted in yellow. By the way,  there is a mistake in the previous report, where 3.1.2 should be 3.1.3.

Sorrr again for the inconvenience! 

Best wishes!

Round 3

Reviewer 3 Report

Comments and Suggestions for Authors

No more comments. The paper can be accepted in the present form.